# The Spatiotemporal Characteristics of Flow–Sediment Relationships in a Hilly Watershed of the Chinese Loess Plateau

**DOI:** 10.3390/ijerph19159089

**Published:** 2022-07-26

**Authors:** Lingling Wang, Wenyi Yao, Peiqing Xiao, Xinxin Hou

**Affiliations:** 1Key Laboratory of the Soil and Water Conservation on the Loess Plateau of Ministry of Water Resources, Yellow River Institute of Hydraulic Research, Yellow River Conservancy Commission, No. 45, Shunhe Road, Zhengzhou 450003, China; wlingling99@163.com (L.W.); wyyaoyrihr@163.com (W.Y.); xinxin_hou@163.com (X.H.); 2Henan Key Laboratory of Yellow Basin Ecological Protection and Restoration, No. 45, Shunhe Road, Zhengzhou 450003, China

**Keywords:** flow–sediment relationship, sediment transportation capacity, topographic unit, different spatial scale, hilly–gully region

## Abstract

The flow–sediment relationship is important to understand soil erosion and sediment transport in severely eroded areas, such as Loess Plateau. Previous research focused on the variation and driving forces of runoff and sediment at the different scales in a watershed. However, the variations of the flow–sediment relationship on multispatial scales (slope, subgully, gully, and watershed scales) and multitemporal scales (annual, flood events, and flood process) were less focused. Taking the Peijiamao watershed, which includes whole slope runoff plot (0.25 ha, slope scale), branch gully (6.9 ha, subgully scale), gully (45 ha, gully scale), and watershed (3930 ha, watershed scale), four different geomorphic units located at the Chinese Loess Plateau, as the research site, a total of 31 flood events from 1986 to 2008 were investigated, and two flood process data were recorded across all the four geomorphic units. The results showed that on the annual timescale, the average sediment transport modulus and runoff depth at four scales exhibited a linear relationship, with determination coefficients of 0.81, 0.72, 0.74, and 0.77, respectively. At the flood event timescale, the relationships between sediment transport modulus and runoff depth at the gully and watershed scales could also be fitted with a linear relationship with high determination coefficients (from 0.77 to 0.99), but the determination coefficient at the slope scale was only 0.37 at the event scale. On the single rainfall event timescale, the flow–sediment relationship at the slope scale showed a figure-eight hysteretic pattern while those relationships at larger scales showed an anticlockwise loop hysteretic pattern. Under the same flow condition, the suspended sediment concentrations during the falling stage were significantly higher than those during the rising stage. Moreover, the difference was bigger as the spatial scale increased due to the wash loads in the downstream gullies, which favored the occurrence of hyper-concentration flow. The results of the study could provide useful insights into the temporal–spatial scale effects of sediment transport and their internal driving mechanisms at the watershed scale.

## 1. Introduction

Runoff and sediment variation and influence factors are one of the hot spots in hydrological science; the flow–sediment relationship is considered as a fundamental element to determine sediment dynamics and there have been numerous studies concerning this topic [1]. Based on the relationship between discharge and suspended sediment concentration, the sediment rating curve could be proposed to determine suspended sediment load [2]. However, the investigation of complicated flow–sediment relationships at multitemporal–multispatial scales could set a robust base for the prediction of sediment yield in ungauged basins.

The frequent soil erosion in the Loess Plateau, which is the most severely eroded area in China and even in the world, presents quite complicated flow–sediment relationships which should be urgently quantified [3]. The Loess Plateau covers over 300,000 km^2^ with diversified landforms. The hilly–gully region covers more than one third of the whole Loess Plateau [4]. Catchments in the hilly–gully region of the Loess Plateau display clear vertical zoning from the top of the slope to the bottom of the valley. The whole slope profile is divided into the hilly slope, gully slope, and the channel [5] (Figure 1c). The area connecting the hilly slope, gully, and valley slope is called the hilly–gully system (also called the whole slope). This symmetric hilly–gully system constitutes the special landform of the Loess Plateau [5,6,7]. Previous research revealed that runoff and sediment yield were highly spatially scale-dependent due to the spatial heterogeneity and non-uniformity [8,9]. Therefore, the flow–sediment relationship may also be spatially scale-dependent and has been rarely studied [3].

Separately, on the slope scale, research has revealed the relationship between runoff and sediment yield in slope gullies of the Loess Plateau. The net erosion of gully slope exhibits a power function relationship with the inflow from the upper hilly slope [10,11]. When the slope runoff increases, there is greater erosion and sediment yield in the gully slope system. The net erosion of the gully slope portion exhibits a negative linear correlation with the sediment concentration; namely, the sediment yield in the gully slope decreases as the suspended sediment concentration in the slope runoff increases. Furthermore, the gravity erosion occurring in the gully slope is the main factor causing the violent change of flow–sediment relationship at the slope scale [12,13,14].

On the watershed scale, previous studies have focused mainly on the small stream and tributary scale and have investigated the spatiotemporal changes of runoff and sediment load and potential flow–sediment relationships [15,16,17,18,19]. Meanwhile, there have been numerous studies concerning the driving forces and their contributions to the variation of flow–sediment relationships [20,21,22,23,24,25]. Different regression models have been developed to quantify the relationship between suspended sediment concentration and runoff. These studies presented valuable insights into the changes in sediment regimes induced by climate change, land use/cover change, and other interventions, such as wildfires and reservoirs [1]. However, few studies have been carried out regarding the spatiotemporal-scale changes of the flow–sediment relationship across different geographic units from slope to watershed in highly eroded regions such as the Loess Plateau.

Thus, in this study, Peijiamao watershed, located in the typical loess hilly and gully region, was selected as the research area, and the long-term (from 1986 to 2008) observation data in representative embedded slope–catchment–gully–watershed prototype observation facilities in the watershed were collected. Based on the hydrometeorological data of each prototype observation facilities, the flow–sediment relationships at four spatial scales (whole slope, subgully, gully, and watershed) and three timescales (annual, flood event, and flood process) for hilly loess areas were examined to better understand the characteristics of runoff and sediment transportation processes from the slope to gully.

## 2. Materials and Methods

### 2.1. Study Area

The study area is located in the first sub-region of the hilly–gully area of the Loess Plateau (Figure 1). The selected watershed (Peijiamao watershed) is an incised watershed, embedded with the subwatershed of Qiaogou gully (gully scale), the first branch of Qiaogou gully (subgully scale), and the entire slope runoff plot of Qiaogou gully (slope scale). The study region is a natural restoration watershed, and there is essentially no human disturbance.

Peijiamao watershed is a primary tributary of the Wuding River and a secondary tributary of the Yellow River. It is located on the left bank of the lower reach of the Wuding River, and the catchment area measures 39.3 km^2^. The elevation difference in the watershed spans 250 m, and the length of the main gully is 11 km. The gully bed slope gradient is 1.51%, the asymmetry coefficient of the watershed is 0.58, and the gully density is 2.69 km/km^2^. Seven rainfall stations and one hydrological station are deployed in the watershed (watershed scale). Qiaogou catchment is the first-order branch of the Peijiamao watershed (gully scale), and the watershed measures 0.45 km^2^. The annual average precipitation in the catchment is approximately 350 mm. The main gully of the watershed is 1.4 km long, and the gully channel slope gradient is 2.11%. The gully density is 5.4 km/km^2^, and the asymmetry coefficient of the watershed is 0.23. The first branch of Qiaogou gully catchment (subgully scale) spans 0.069 km^2^, the gully length is 870 m, and the gully channel gradient is 4.97%. In the catchment of Qiaogou gully, four rainfall stations and three hydrological stations are deployed (Figure 1d). The entire-slope runoff field (slope scale) is located on the left bank slope of the Qiaogou gully catchment, where the average slope gradient is 32°18′, the inclined slope length is 117 m, the horizontal slope length is 98.9 m, and the average width is 25.2 m. The horizontal area of the runoff field measures 2492 m^2^, and the inclined area measures 2948 m^2^. The terrain characteristics of the geomorphic units of different spatial scales are shown in Table 1.

### 2.2. Prototype Observation Facility

The rainfall stations in the research area were all equipped with automatic rainfall gauge (DSJ2-type hydro-cone) for recording rainfall dynamic and amounts. Peijiamao watershed (representing watershed scale), Qiaogou gully (representing gully scale), and the first branch of Qiaogou gully (representing subgully scale) were instrumented with triangular weirs for recording runoff and collecting sediment samples. At the outlet of the entire-slope runoff field (representing slope scale), the triangular measuring tank was installed. The recorded data included the flow rate and sediment concentration for each rainfall event, and soil water contents before and after rainfall event were also measured.

### 2.3. Data Collection and Processing

The data of rainfall, flow rate, and sediment concentration in the whole slope runoff field, the first branch of Qiaogou gully, Qiaogou gully, and Peijiamao watershed were all collected from the Suide Soil and Water Conservation Experimental Station of the Yellow River Conservancy Commission during the period of 1986–2008. For 31 flood events, flow rate and sediment yield were compared. For two other flood events (Table 2), the detailed hydrological processes were simultaneously monitored across different spatial scales (all the four geomorphic units).

The major equations used in this study are given as follows:(1)$$W_{a}=\sum_{i=1}^{365}Q_{i}\times 24\times 60$$
(2)$$h_{a}=\frac{W_{a}}{1000\times A\times 10^{6}}$$
where Wa is annual discharge (m^3^ a^−1^); *Q_i_* is discharge (m^3^ s^−1^) at *i*th spatial scale during measuring time *t* (second); $h_{a}$ is annual average runoff depth (mm); *A* is watershed area (km^2^).
(3)$$S{ed}_{i}=\sum_{0}^{T}Q_{i}\times SSC_{i}$$
(4)$$Sed_{a}=\sum_{i=1}^{365}SSC_{i}\times 24\times 60$$
(5)$$SDM_{a}=\frac{Sed_{a}}{1000\times A}$$
where *Sed_i_* is total sediment yield (kg T^−1^) at *i*th spatial scale during the certain timescale (annual or event); *SSC_i_* is sediment concentration (kg m^−3^) at *i*th spatial scale at sampling time; *Sed_i_* is the annual sediment yield (kg a^−1^); $SDM_{a}$ is annual sediment transport modulus, t km^−2^.

Furthermore, ArcGIS 9.2 software (Beijing ESRI company, Beijing, China) was used to interpolate the rainfall map in the Qiaogou catchment. Office 2013 professional (Henan huichuangjiahe science and Trade Co., Ltd., Zhengzhou, China) was used to process and analyze the data. 

## 3. Results and Analysis

### 3.1. Flow–Sediment Relationship at the Annual Timescale

The annual average runoff depth and sediment transport modulus of the different geomorphic units reflected the total annual runoff and sediment yield at different scales. A scatter plot of the relationships between the annual runoff depth and sediment transport modulus of different geomorphic units is illustrated in Figure 2. The correlation analysis shows that these annual runoff depths and sediment transport moduli were significantly correlated at the *p* < 0.01 level. Figure 2 shows that the relationships between the total amounts of runoff and sediment concentration of the geomorphic units at different spatial scales could be fairly well fitted with a linear function (*p* < 0.01). The determination coefficient (*R*^2^) of the functions at the whole slope, Qiaogou subgully, Qiaogou gully, and Peijiamao watershed were 0.81, 0.72, 0.74, and 0.81, respectively. Moreover, the linear slope of the flow–sediment relationships of the different geomorphic units ranged from 169–321. Therefore, the flow–sediment relationship for the slope and watershed scale was generally similar on the annul timescale, with the slopes higher at small scales than at larger scales.

### 3.2. Flow–Sediment Relationships at Flood Events Timescale

The linear relationships between runoff depth and sediment transport modulus for the different geomorphic units across four spatial scales during 31 flood events are plotted in Figure 3. The difference between the flow–sediment relationship on the slope scale and watershed scale is relatively large. As the spatial scale increases, the distribution of runoff depth and sediment transport modulus is more concentrated to the fitted curve, showing more heterogeneity of sediment transport at small scales. The determination coefficient increased from 0.37 to 0.99, indicating that as the spatial scale increased, the relationship between the runoff depth and sediment transport modulus was more robust. It meant that the larger the basin scale was, the more stable the flow–sediment relationship appeared. Therefore, on the basin scale, the amount of sediment could be predicted based on the stable relationships during flood events. Zheng et al. established a linear regression function between the sediment transport modules and runoff depth in different geomorphic units based on different periods of historical dataset, and the slopes of the linear regression between the sediment transport modulus and runoff depth at different scales ranged between 475 and 803, which indicated that the flow–sediment relationship had similarity at different scales on flood event scale, with the lowest slope at watershed scale, indicating stable spatially-averaged low sediment yields at larger scale [26].

Further, the lower determination coefficient was only 0.37 for the runoff plot of the whole slope, which indicated that on the timescale of flood event, the variation of the flow–sediment relationship at slope scale was larger than other three scales. Due to the randomness and uncertainty of the occurrence of gravity erosion at this scale, sediment yield varied significantly with a given amount of precipitation runoff, which resulted in a relatively large variation in the flow–sediment relationship.

In order to compare with other studies which focused on the flow–sediment relationship at flood event [27], the discharge–sediment concentration relationship at different scales (Figure 4) was also illustrated based on a total of 31 flood events during the 23 years. It is noted that the average discharge and suspended sediment concentration at slope scale (whole slope plot) was fitted (*p* < 0.05) by a power function (Figure 4d), while they were also significantly fitted (*p* < 0.01) at larger spatial scales (subgully, gully, and Peijiamao watershed (Figure 4a–c). Furthermore, as the scale increased, the determination coefficient of the power function increased. However, when the discharge was greater than a certain value, the suspended sediment concentration approached a certain constant value. This pattern was likely related to the fact that hyper-concentrated flows were common in the loess hilly region [3,5,28,29].

### 3.3. Flow–Sediment Relationships’ Evolution at the Single Rainfall Event Timescale

As is illustrated in Figure 4, the relationship of discharge and suspended sediment concentration varies significantly with spatial scale increasing, which is a prominent feature of runoff and sediment yield process in loess hilly and gully region [30]. The two floods, on 18 September 2008 and 19 July 2009 (referred to as floods 080918 and 090719), were selected in this study to show flow–sediment evolution processes during floods and their scale effects. The event on 18 September 2008 was generated by a rainfall of 27.3 mm with a mean rainfall intensity of 13.65 mm/h. The rainfall duration is short and the rainfall intensity is high (Table 2), and the rainfall center occurs in the lower reaches of Qiaogou gully (Figure 5). The event on 19 July 2009 was generated by a rainfall of 49.2 mm with a mean rainfall intensity of 7.02 mm/h. The total amount of rainfall is large, but the intensity is lower, and the rainfall center occurs in the upper reaches of the Qiaogou gully (Figure 5). As there was no flooding at the Peijiamao outlet station during the two events, only the flow–sediment relationships in Qiaogou gully were analyzed. Figure 6a,b show the complex hysteresis at the different spatial scales in Qiaogou gully.

The flow–sediment relationship at different spatial scales showed the different hysteretic pattern during two events. At slope scale, the flow–sediment relationship exhibits a figure-eight hysteretic pattern which has two or more loops (Figure 6a). Otherwise, at larger scales, the flow–sediment relationship showed the anticlockwise hysteretic pattern, which was characterized by a delayed increase in suspended sediment concentration (Figure 7). It can be inferred that sediment transported from remote areas entered the outlet stations when flood peaks elapsed. Furthermore, there is no significant difference in suspended sediment concentration at the same discharge level in the rising and falling stage of a flood event for the slope scale. However, for the larger scales, with the increasing of spatial scale, the lower half of the anticlockwise loop corresponded to the rising stages till discharge reached peaks and sediment concentration reached peaks, and the upper half corresponded to the falling stages following the discharge peaks and sediment concentration peaks. Given a constant discharge, the sediment concentration during the falling stage was significantly higher than that during the rising stage. The difference between the sediment concentrations during the two stages was greater at larger scales. The second flushes during the two rainfall events might trigger higher SSC values. The hyper-concentration flow triggered by gully erosion might contribute to this hysteresis phenomenon. These flow–sediment relationships confirmed that runoff and sediment production in the loess hilly and gully region were unique both in temporal and spatial scales [30].

## 4. Discussion

In this study, we compared the relationships of *SDM*–*h* and *SSC*–*Q* on the annual timescale, multiple events timescale, and the single rainfall event at different spatial scales across the Peijiamao watershed, which is located in the Loess Plateau. The results show that the flow–sediment relationships presented distinct scale effects, with more frequent hyper-concentration flow occurring at larger scales. It is suggested that the local concentration limit or the maximum transport capacity approached due to the gully erosion which triggered soil detachment in gullies and can provide sufficient sediment in the Loess Plateau [31]. When the runoff reached a state with high sediment concentration, its high sediment-carrying capability could be maintained even by a relatively small flow rate. Therefore, when the spatial scale was up to the watershed, the flow–sediment relationship showed spatial invariance both on the annual timescale and the flood event timescale. These conclusions are consistent with the results of Zheng et al [3]. The results support the hypothesis of Asselman that flat rating curves should be characteristic for river sections where easily erodible materials [15], such as weathered materials or loose sedimentary deposits, are sufficiently available at almost all discharges. However, at the slope scale, it is difficult to fit the flow–sediment relationship with linear function on multiple flood event timescales. These phenomena support the hypothesis of Asselman that the watershed characteristics shaped the sediment rating curves [15]. However, during the single rainfall event, the different relationships of *SSC*–*Q* between slope scale and watershed scale, along with different hysteretic patterns, further explained that the sediment peak was induced by hyper-concentration flow mechanism in downstream gullies. The second flushes during the two rainfall events strengthened the washing load process, which caused higher SSC concentrations.

For the flow–sediment relationship on the single rainfall event, the spatial and temporal variations can result in changes in the sediment sources and are responsible for different hysteretic loop patterns [32]. At the slope scale, due to the steep slope, the runoff generation time and concentration time are very short, and the flood regulation ability is poor. Furthermore, the upper runoff and sediment from the hilly slope have a great impact on the soil erosion of the lower gully slope [3]. The scale effect on soil erosion at slope scale is variable and dynamic [12,33,34]; however, at larger scales, the presence of base flow was assumed to be responsible for the change of the flow–sediment relationship, which favored the occurrence of hyper-concentration flow. The influence of upstream sediment-laden flow on downstream output was limited, and spatial scale effects on sediment-laden flow decreased with the increasing drainage area, especially for major sediment-producing events with area-specific sediment yield larger than 300 t/km^2^ [31,35].

## 5. Conclusions

Based on the long-term runoff and sediment data during the years of 1986–2008 across scales from a whole slope runoff plot, for the first branch of Qiaogou (subgully), Qiaogou gully to Peijiamao watershed, a total of 31 flood events and 2 single rainfall events with detailed synchronous monitoring of runoff and sediment were selected to analyze the flow–sediment relationships at the timescales of annual timescale, multiple flood events timescale, and single rainfall event. The conclusions are as follows:

(1)On the annual timescale, the flow–sediment relationship was stable. The annual runoff depths and annual sediment transport modulus exhibited linear relationships, and the determination coefficients ranged from 0.72 to 0.81 (*p* < 0.01). The differences between the flow–sediment relationships of different geomorphic units across scales were not significant (*p* > 0.05).(2)On the multiple flood events timescale, the sediment transport modulus and runoff depth in watershed spatial scale could also be fitted with linear relationships, with the high determination coefficients from 0.37 to 0.99 (*p* < 0.05), and the suspended sediment concentration and discharge was fitted by power functions with determination coefficients greater than 0.22 (*p* < 0.05), and especially higher (*R*^2^ > 0.52, *p* < 0.01) at larger scales. The flow–sediment relationship at the slope scale changed violently due to the random gravity erosion that happened.(3)On the single rainfall event timescale, the flow–sediment relationships showed different hysteretic patterns across different spatial scales. The figure-eight hysteretic pattern was present at the slope scale while the anticlockwise hysteretic pattern was present at the subgully and gully scales. Furthermore, at a given discharge, the suspended sediment concentration during the falling stage was higher than that during the rising stage. Moreover, the difference was bigger as the spatial scale increased due to the wash load in the gullies, which favored the occurrence of hyper-concentration flow.

The results of this study could provide useful insight for understanding the flow–sediment relationship, which can serve as a reference for soil conservation planning in other similar regions.

## Figures and Tables

**Figure 1 ijerph-19-09089-f001:**
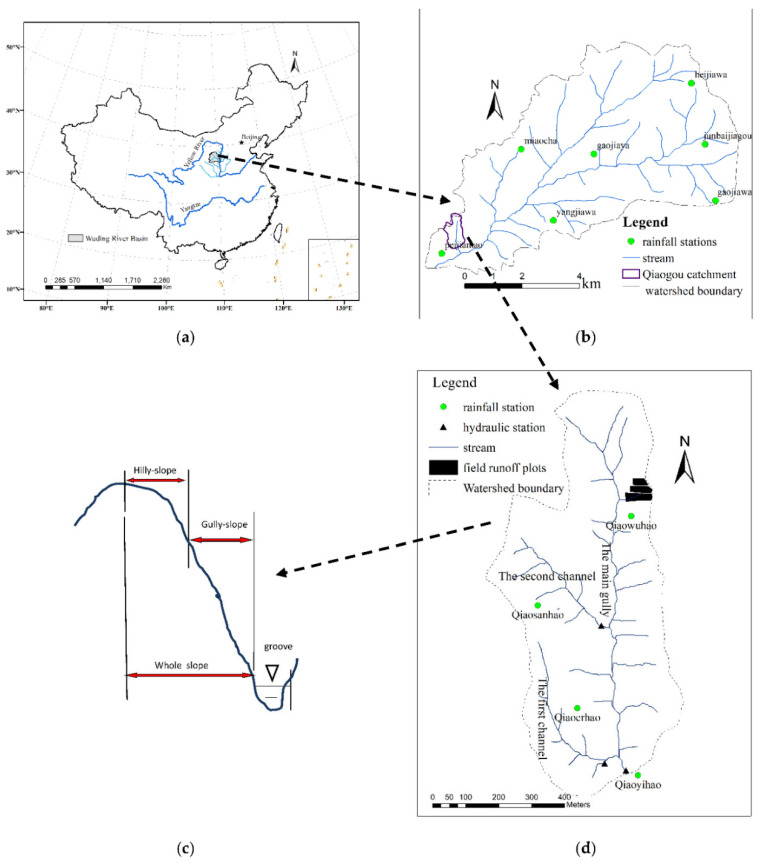
Location of the study area and the instrumentation in the watershed. (**a**) Reaches of the Yellow River. (**b**) Peijiamao watershed. (**c**) Profile of the runoff plot. (**d**) Qiaogou catchment.

**Figure 2 ijerph-19-09089-f002:**
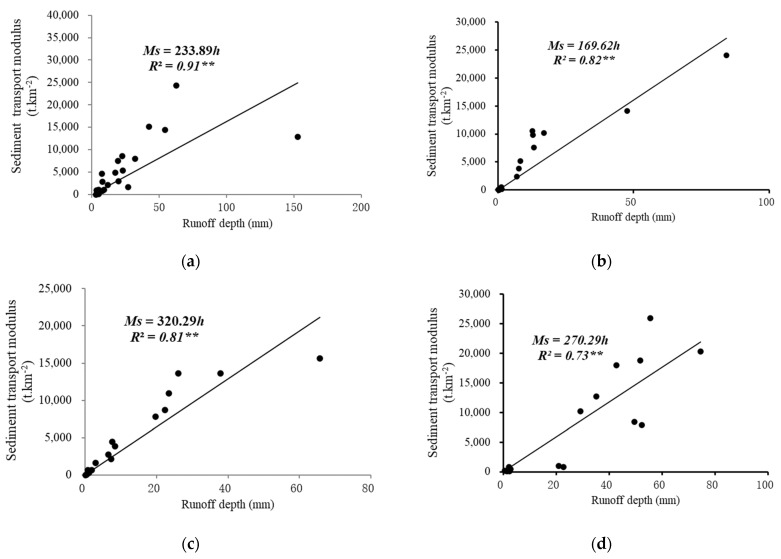
Relationships between runoff depth and sediment transport modulus of the different geomorphic units on annual scale. (**a**) Peijiamao watershed; (**b**) Qiaogou gully; (**c**) first branch of Qiaogou gully (subgully); (**d**) whole slope. Note: ** significance at the 0.01 level; *n* = 18.

**Figure 3 ijerph-19-09089-f003:**
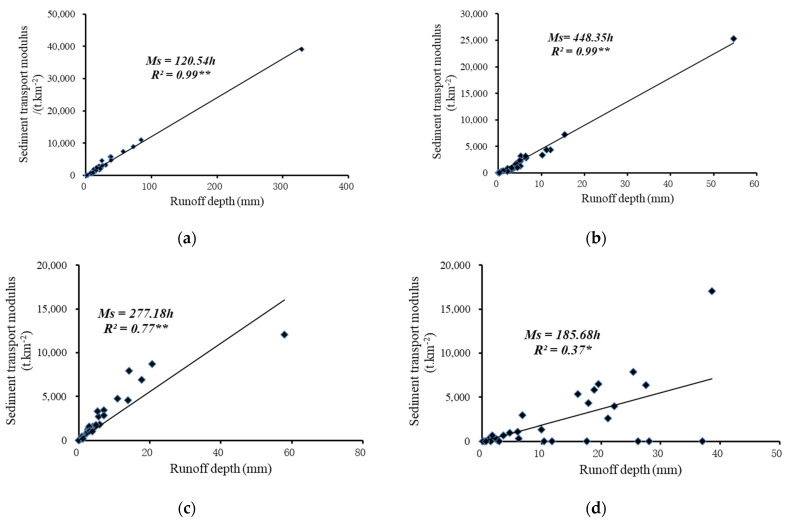
Relationships between runoff depth and sediment transport modulus on flood event scale. (**a**) Peijiamao watershed; (**b**) Qiaogou gully; (**c**) first branch of Qiaogou gully (subgully); (**d**) whole slope. Note: ** significance at the 0.01 level; * significance at the 0.05 level; *n* = 31.

**Figure 4 ijerph-19-09089-f004:**
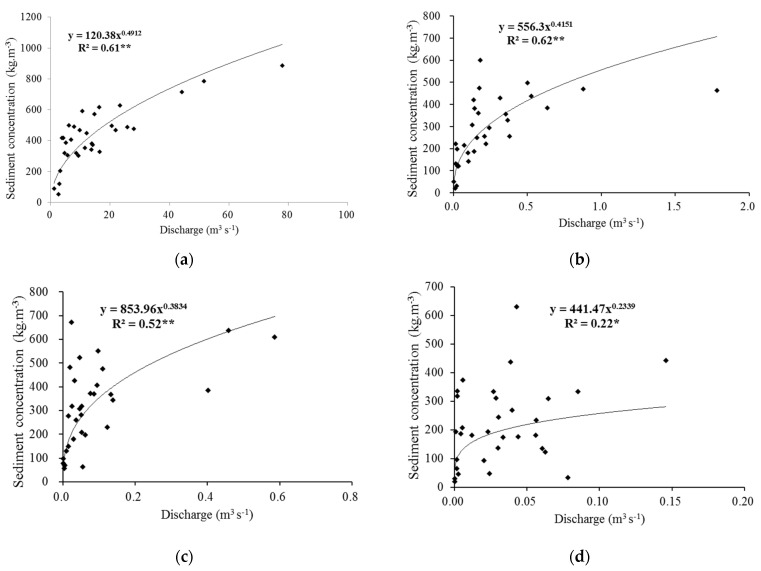
SSC–Q relationships of different geomorphic units at flood event timescale. (**a**) Peijiamao watershed; (**b**) Qiaogou gully; (**c**) first branch of Qiaogou gully (subgully); (**d**) whole slope. Note: ** significance at the 0.01 level; * significance at the 0.05 level; *n* = 31.

**Figure 5 ijerph-19-09089-f005:**
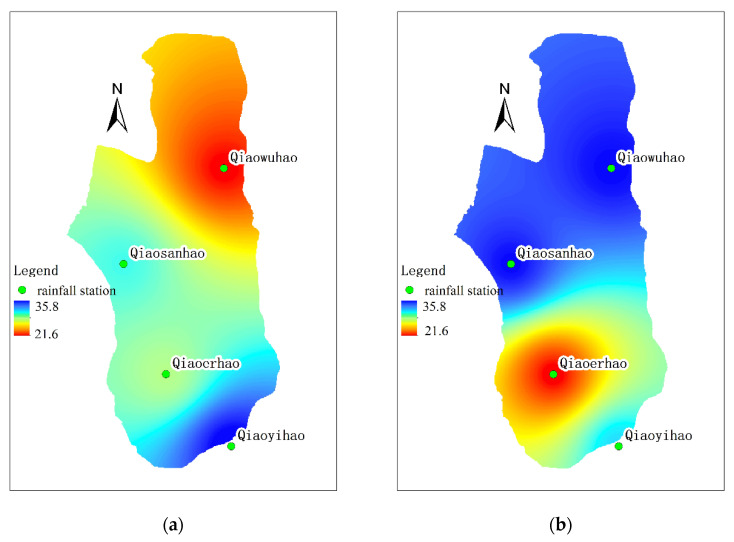
Rainfall spatial distribution on 18 September 2008 and 19 July 2009 in the Qiaogou watershed: (**a**) 18 September 2008; (**b**) 19 July 2009.

**Figure 6 ijerph-19-09089-f006:**
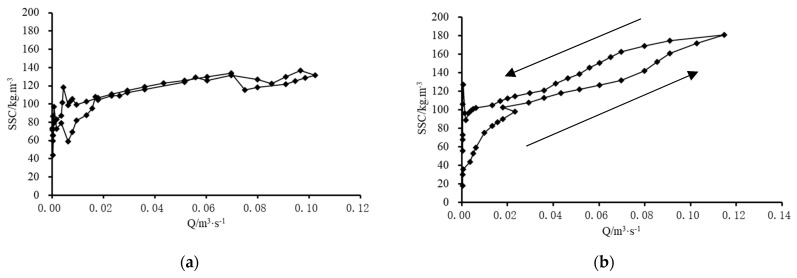
Relationships between SSC and Q of the different geomorphic units on 18 September 2008. (**a**) Whole slope; (**b**) first branch of Qiaogou (subgully); (**c**) Qiaogou gully.

**Figure 7 ijerph-19-09089-f007:**
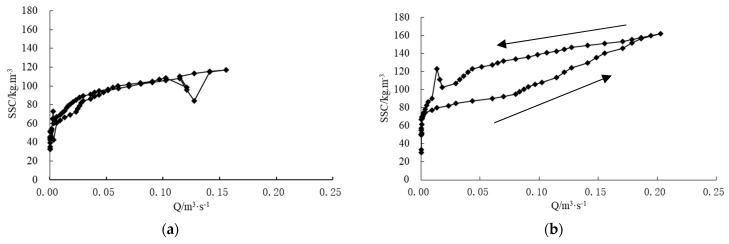
Relationships between SSC and Q of the geomorphic units on 19 July 2009. (**a**) Whole slope; (**b**) first branch of Qiaogou (subgully); (**c**) Qiaogou gully.

**Table 1 ijerph-19-09089-t001:** Description of field runoff plot and gauging watershed.

No.	Geomorphic Unit (Scale)	Area, km^2^	Length, km	Slope, °	Gully Gradient, %	Basin Axis Direction
1	Whole slope (slope)	0.0025	0.177	32.3	-	West
2	First branch (subgully)	0.069	0.870	-	4.97%	Northwest–southeast
3	Qiaogou gully (gully)	0.45	1.400	-	2.11%	North
4	Peijiamao watershed (watershed)	39.3	11.00	-	1.51%	Northeast–southwest

**Table 2 ijerph-19-09089-t002:** Rainfall conditions of two flood events.

Floods	Start Time of Rainfall			End Time of Rainfall		Rainfall at Each Station				Average Rainfall
	Month	Day	Hour	Day	Hour	Q_1_	Q_2_	Q_3_	Q_5_	
20080918	9	18	16:05	18	18:10	35.9	28.5	30.5	21.6	27.3
20090719	7	19	17:20	20	1:35	49.3	47.8	49.9	49.9	49.2

## Data Availability

Not applicable.

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
