# Peer review of "The Spatiotemporal Characteristics of Flow–Sediment Relationships in a Hilly Watershed of the Chinese Loess Plateau"

_ijerph, 2022, doi:10.3390/ijerph19159089_

Round 1

Reviewer 1 Report

Dear Authors,

The authors presented the spatial and temporal characteristics of flow-sediment relationships in an eroded area. The article has valuable content for the Special Issue's Journal's audience. The article is simple but concise. The strength of the paper is in the merge of the scientific overview into the practical sphere of the Chinese Loess Plateau. The references can use some work. The statements are coherent and supported by listed citations. Figures are generally ok, but one presented (Figure 5) should be improved.

A personal comment: When sending such a simple article to a journal that is ranked Q1 in "PUBLIC, ENVIRONMENTAL & OCCUPATIONAL HEALTH", and IF over 3, a decent thing would be to edit the article accordingly before sending it out, or in other words, to read it through a couple of times by the authors. The paper has multiple text flaws and errors and I won't be going through them, due to too many of them. Saying mentioned, expect them to be corrected fully and don't believe the reviewer's job is to point them out to you (honest mistakes happen, but this is clearly neglect). Even though I am frustrated while reading, I will mark the article in an objective manner.

33% (out of only 36 cited papers) are "In Chinese", while 16.7 % of all cited papers, that are in English, are from the last 5 years. This should be improved. This would be ok for some kind of a technical paper, but not a research paper like this. Saying the mentioned, the presented references are used in a good manner.

My biggest criticism is probably for the Introduction. Overall, the intro is too short and basic. Please expand thoughts a bit outside of the Loess Plateau as well. The study area, observation facility, and data are well described. Results seem to be alright. Discussion is written in a good manner. Conclusions are satisfactory, useful to this community, and well described.

I will be pushing this paper forward due to the suitability of the Special Issue.

L36 - Every problem is important. Please rephrase.

L42 - Please put "whole slope" under brackets or rephrase, this sentence is very banal.

L189 - You have already stated what a "whole slope" is, no need to repeat

L241 - Figure 5, please improve the quality of the text in the figure, it is not clearly visible.

L245 - Figure 6 and Figure 7 - when comparing different spatial scales, locations, depths, etc, it is useful if you make the graphs (X & Y axis) directly comparable. In other words, Figure 6: X -> max 0.15 and Y ->200 and Figure 7 > max 0.25 and Y ->180. Or something similar.

Author Response

Response to the Comments of the Reviewer 1

Comment 1

The authors presented the spatial and temporal characteristics of flow-sediment relationships in an eroded area. The article has valuable content for the Special Issue's Journal's audience. The article is simple but concise. The strength of the paper is in the merge of the scientific overview into the practical sphere of the Chinese Loess Plateau. The references can use some work. The statements are coherent and supported by listed citations. Figures are generally ok, but one presented (Figure 5) should be improved.

A personal comment: When sending such a simple article to a journal that is ranked Q1 in "PUBLIC, ENVIRONMENTAL & OCCUPATIONAL HEALTH", and IF over 3, a decent thing would be to edit the article accordingly before sending it out, or in other words, to read it through a couple of times by the authors. The paper has multiple text flaws and errors and I won't be going through them, due to too many of them. Saying mentioned, expect them to be corrected fully and don't believe the reviewer's job is to point them out to you (honest mistakes happen, but this is clearly neglect). Even though I am frustrated while reading, I will mark the article in an objective manner.

33% (out of only 36 cited papers) are "In Chinese", while 16.7 % of all cited papers, that are in English, are from the last 5 years. This should be improved. This would be ok for some kind of a technical paper, but not a research paper like this. Saying the mentioned, the presented references are used in a good manner.

My biggest criticism is probably for the Introduction. Overall, the intro is too short and basic. Please expand thoughts a bit outside of the Loess Plateau as well. The study area, observation facility, and data are well described. Results seem to be alright. Discussion is written in a good manner. Conclusions are satisfactory, useful to this community, and well described.

Response:

The authors are grateful to the reviewer for his constructive comments and suggestions. The manuscript had been revised thoroughly, both in scientific contents and writing. 

The Figure 5 was redrawn to improve the quality.

The introduction part had been supplemented with latest literature and link the research to a broader context (Lines 39-51, 57-61, 72-84).

All the changes had been marked with red in the manuscript.

Comment 2

L36 - Every problem is important. Please rephrase.

Response:

We had rephrased the whole introduction part, especially Lines 39-51, 57-61, 72-84.

Comment 3

L42 - Please put "whole slope" under brackets or rephrase, this sentence is very banal.

Response:

The spatial terms had been totally reorganized through the manuscript.

Comment 4

L189 - You have already stated what a "whole slope" is, no need to repeat

Response:

The repeated parts had been deleted in the revised manuscript.

Comment 5

L241 - Figure 5, please improve the quality of the text in the figure, it is not clearly visible.

Response:

The Figure 5 was redrawn to improve the quality.

Comment 6

L245 - Figure 6 and Figure 7 - when comparing different spatial scales, locations, depths, etc, it is useful if you make the graphs (X & Y axis) directly comparable. In other words, Figure 6: X -> max 0.15 and Y ->200 and Figure 7 > max 0.25 and Y ->180. Or something similar.

Response:

We fully agree with the reviewer’s suggestion. The Figure 6 and 7 were modified to be more comparable.

Reviewer 2 Report

The manuscripts present the results of a field experimental study to assess the effects of temporal and spatial scales on the flow-sediment relationships. The work could have been in such a way to be more understandable and useful to the readers. It lacks enough information on the experimental, setuphow the data have been collected, and the accuracy of the data. The data processing is also too brief without providing any details. Considering these two issues, it is too difficult to evaluate the results of this work.

Author Response

Response to the Comments of the Reviewer 2

The manuscripts present the results of a field experimental study to assess the effects of temporal and spatial scales on the flow-sediment relationships. The work could have been in such a way to be more understandable and useful to the readers. It lacks enough information on the experimental, setup how the data have been collected, and the accuracy of the data. The data processing is also too brief without providing any details. Considering these two issues, it is too difficult to evaluate the results of this work.

Response:

The experimental setup had been rewritten, the data collection and processing procedure were detailedly introduced (Lines 141-176).

The whole manuscript had been thoroughly revised to improve the logic lines and to highlight the contributions to the communities.

All the changes had been marked with red in the manuscript.

Round 2

Reviewer 2 Report

The manuscript is improved significantly compared to the first draft. However, it still required very careful proofreading since there are a lot of places with English grammer or sentence structure problem. I have highlighted a few of them in the attached file.

Author Response

Line-39

Do you mean "hot topics" or "hot research area"?

Response:

The hot spots in the paper refer to the "hot topics".

Line-39

What do you mean by influence factors?

Response:

  The influence factors of runoff and sediment variation include underlying surface changes and human activities, such as vegetation coverage, land use,  implemented water and soil conservation measures, etc.

Line-49

More than?

Response:

  Yes, we missed the word “than”.

Line-49

Inverse?

Response:

 Yes. There is a negative correlation between the net erosion of gully slope and the sediment concentration of the hilly slope.

Line-196

which parameters does this note refer to?

Response:

Here, the parameter is the decisive coefficient R2.

Line-260

Low?

Response:

Yes. the low is more accurate. So, the word “slow” changes to the lower.

Line-311

Show?

Response:

Yes.